# Depressive Symptoms and Risk Factors During the COVID-19 Pandemic Among People with/Without Mental Disorders

**DOI:** 10.3390/healthcare13101095

**Published:** 2025-05-08

**Authors:** Yuko Fukase, Kanako Ichikura, Hidenori Inaoka, Hirokuni Tagaya

**Affiliations:** School of Allied Health Sciences, Kitasato University, 1-15-1, Kitazato, Minami-ku, Sagamihara 252-0303, Kanagawa, Japan; ichikura@kitasato-u.ac.jp (K.I.); inaoka@kitasato-u.ac.jp (H.I.); tagaya@kitasato-u.ac.jp (H.T.)

**Keywords:** mental disorders, COVID-19 pandemic, depressive symptoms, risk factors, coping

## Abstract

**Background/Objectives**: The present study aimed to reveal the differences in changes in and risk factors for depressive symptoms between people with and without various psychiatric disorders during the pandemic. **Methods**: Longitudinal web-based surveys were conducted from 2020 to 2022. The diagnosis of mental disorders was based on self-reports by participants. Depressive symptoms were measured via the Patient Health Questionnaire-9 (PHQ-9), and coping was measured via the Brief Coping Orientation to Problems Experienced Inventory. A linear mixed model of PHQ-9, two-sample *t*-tests on Brief-COPE, and multiple linear regression for with and without mental disorders were conducted. **Results**: A total of 1443 participants were analyzed, of whom 9.3% had mental disorders. Depressive symptoms significantly decreased from January 2021 to January 2022, regardless of mental disorder status. Participants with mental disorders used certain coping styles more frequently than those without mental disorders. In a multiple linear regression analysis, no coping strategy was significantly effective for PHQ-9 scores among participants with mental disorders. However, being single was a risk factor, and emotional support use was associated with PHQ-9 scores. Additionally, behavioral disengagement was linked to PHQ-9 scores, regardless of mental disorder status. **Conclusions**: These results showed depressive symptoms might decrease in the long term regardless of the presence of mental disorders. Although there was no evidence of coping strategies effectively reducing depressive symptoms in people with mental disorders, the presence of a spousal relationship may play an important protective role for people with mental disorders and behavioral guidelines regardless of the presence of mental disorders.

## 1. Introduction

The coronavirus disease 2019 (COVID-19) pandemic threatened people’s mental health worldwide [1,2,3]. A systematic review reported that the prevalence of depressive disorders increased during the COVID-19 pandemic compared to before the pandemic [1]. COVID-19 Mental Disorders Collaborators [1] reported an additional 53.2 million cases of major depressive disorder globally in 2020 due to the effects of the COVID-19 pandemic, noting that the increased prevalence of depressive disorders was associated with higher SARS-CoV-2 (severe acute respiratory syndrome coronavirus 2) infection rates and decreasing human mobility.

Previous studies have reported that having a mental disorder is a risk factor for worsening depressive symptoms and overall mental health deterioration during the pandemic [4,5,6,7,8,9]. People with mental disorders might continue to be vulnerable to emotional distress up to one year after the start of a pandemic [10]; highlighting the need for long-term studies, although most existing research has been short-term.

Coping strategies are cognitive and behavioral efforts used to manage mental symptoms. Lazarus [11] reported that coping strategies might play a role as mediators of the stress—illness relationship. People with mental disorders tended to use passive and avoidance coping styles during the COVID-19 pandemic [4,12]. However, these surveys were cross-sectional and conducted in 2020, and the long-term effects of coping styles on mental health in people with mental disorders could not be investigated.

The aim of the present study was to investigate depressive symptoms, risk factors, and coping styles during the COVID-19 pandemic among people with and without mental disorders. We conducted a longitudinal study from 2020 to 2022 in the general population in Japan. The participants were divided into groups with and without mental disorders.

This study examined changes in depressive symptoms among people with mental disorders from 2020 to 2022 and compared these changes with those in people without mental disorders. Differences in coping styles for managing pandemic-related social changes and inconveniences were also explored between the two groups. Furthermore, we compared risk factors and effective coping strategies for depressive symptoms between people with and without mental disorders.

In the present study, we categorized psychiatric disorders under the broad term “mental disorder”, rather than disaggregating by specific diagnoses. This decision was informed by previous research highlighting shared psychological vulnerabilities and common stress response patterns across a wide spectrum of psychiatric conditions [13,14,15]. By analyzing these conditions collectively, we aimed to identify overarching trends that may be generalizable beyond diagnostic boundaries, thereby contributing to a broader understanding of the experiences of people with mental health challenges.

## 2. Materials and Methods

### 2.1. Procedure

A total of 6 times web-based surveys were conducted during the COVID-19 pandemic [16,17,18]. The survey periods, selection criteria, and a quota sampling method are shown in Figure 1. The survey period was dependent on resources such as research funds, so there was no regularity in the timing of the surveys.

In the present study, the longitudinal data from September 2020 (Time 1) to July 2022 (Time 5) were analyzed because the data of diagnosis of mental disorder was measured from the survey conducted at Time 1.

Participants were recruited from a pool of approximately 10 million individuals registered with an online research company Macromill, Inc., Tokyo, Japan and from companies with which Macromill has partnerships. All the participants received Macromill points for their participation; Macromill points are associated with the original point service provided by Macromill, Inc., and individuals can trade these points for prizes or cash.

The Research Ethics Review Committee of Kitasato University School of Allied Health Sciences approved this study (the approval numbers were 2020–011 on 13 July 2020 and 2020–023 on 14 September 2020). All participants were informed of the aims of the present study and their right to cease participating in the survey prior to their participation. For participants, checking a box indicating “I agree to participate in the study” was considered to be an indication of consent, and informed consent was obtained from all participants.

### 2.2. Measures

Data on sociodemographic characteristics, diagnosis of the presence or absence of mental disorders, depressive symptoms, and coping style used, were collected in every survey from Time 1 to Time 5.

The sociodemographic characteristics and diagnosis of mental disorders were collected in the survey at Time 1. The sociodemographic characteristics investigated included age, sex, marital status, the presence or absence of children, employment status, household income level (i.e., <2 million JPY as a low-income level, 2–8 million JPY as a middle-income level, and >8 million JPY as a high-income level), and the economic impact of the COVID-19 pandemic.

The diagnosis of mental disorders was based on self-reporting by the participants. The participants responded to multiple-choice items regarding whether they had been diagnosed with any of the following disorders: schizophrenia, depression, bipolar disorder, stress-related and somatoform disorders, neurotic disorders, and dementia. These items were selected from the Patient Survey, one of the statistical surveys conducted by the Ministry of Health, Labor, and Welfare of Japan, which is based on the 10th revision of the International Statistical Classification of Diseases (ICD-10).

Depressive symptoms were collected in every survey from Time 1 to Time 5 and were measured via the Japanese version of the Patient Health Questionnaire-9 (PHQ-9) [19,20]. The participants were asked to indicate the frequency with which they had experienced depressive symptoms over the past 2 weeks. The PHQ-9 consists of 9 items scored on a four-point scale (0 to 3); the total score ranges from 0 to 27 points, with higher scores indicating more depressive symptoms. The Japanese version of the PHQ-9 has been validated for depression screening in primary care settings, and the relationship of the PHQ-9 score with the Short-Form 8 (SF-8) mental component summary score was reported to be significant [20].

Coping strategies were assessed at each time point from Time 1 to Time 5 using the Japanese version of the Brief Coping Orientation to Problems Experienced (Brief-COPE) Inventory [21,22]. This scale evaluates 14 distinct coping styles and allows for a more detailed classification into categories such as problem-focused and emotion-focused coping [22]: self-distraction, active coping, denial, substance use, emotional support, instrumental support, behavioral disengagement, venting, positive reframing, planning, humor, acceptance, religion, and Self-blaming. Participants in the present study were asked to report the frequency with which they employed various coping methods to handle the social disruptions and challenges arising from the COVID-19 pandemic at the time of completing each survey. The scale consists of 28 items, with each coping style assessed using two items. Each item is rated on a 4-point scale (1 to 4), yielding a total score ranging from 2 to 8 for each coping style. A higher score reflects more frequent use of that particular coping style. The Japanese version of the Brief-COPE has been validated for use among workers in Japan, with significant correlations reported between coping styles and negative emotions, fatigue, concentration, and activity levels [22].

### 2.3. Sample Size

The first study of the survey at Time 0 aimed to recruit 2700 participants and ultimately recruited 2708 participants [16]. In the first study, the calculation of the appropriate sample size was based on the following criteria: the initial study incorporated 44 independent variables in the logistic regression analysis, and the required sample size was calculated to be 10 times the number of these variables [23]. Additionally, the first study planned to conduct the analysis using a probable depression group selected from the participants, which was based on a PHQ-9 score of 10 or higher. Previous research reported that the prevalence of depressive symptoms during the COVID-19 pandemic was likely less than 17%. As a result, the initial study needed over 440 participants to be classified into the group with probable depression.

In contrast, the current study was longitudinal, and it was not possible to manage the sample size. Although the sample size of participants with mental disorders was expected to be small, unavoidable limitations in sample size due to resource constraints exist, and validity could not be determined based on this factor alone [24]. This issue will be discussed in Section 4.

### 2.4. Statistical Analysis

Descriptive statistics were calculated for the sociodemographic characteristics at Time 1. The chi-square test and two-sample *t* test were conducted among participants with and without mental disorders.

A linear mixed model of PHQ-9 scores was constructed. The fixed effects were repeated scores from Times 1 to 5 and the scores of participants with and without mental disorders, and the random effect was individual differences.

For the Brief-COPE Inventory score at Time 5, two-sample *t*-tests were conducted between participants with and without mental disorders. Welch’s *t*-test was conducted when the values for participants with and without mental disorders were not equally distributed.

Multiple linear regression for participants with and without mental disorders was subsequently conducted. The PHQ-9 score at Time 5 was the dependent variable, the PHQ-9 score at Time 1 was the adjustment variable, and the sociodemographic characteristics at Time 1 and the Brief-COPE Inventory score at Time 5 were the predictor variables.

Pearson’s product–moment correlation coefficient was calculated to check for collinearity and validate relationships between variables. If this value exceeded 0.8, the variable was excluded from the multiple linear regression analysis. Similarly, if the Variance Inflation Factor (VIF) value exceeded 5.0, the variable was excluded from the multiple linear regression analysis.

The statistical significance level was set at *p* < 0.05. All statistical analyses were carried out via IBM SPSS Statistics (Version 28).

## 3. Results

### 3.1. Characteristics of the Participants

Figure 2 provides an in-depth overview of the participant selection process. At Time 0, 2708 individuals took part in the survey. Of these, 595 participants did not participate in the survey at Time 1, and 670 participants did not participate in the survey at Time 5; accordingly, 1443 participants were included in the present study (response rate of 53.3%). Among the included participants, 134 (9.3%) reported having a diagnosis of mental disorders, while 1309 (90.7%) reported not having any mental disorders.

The sociodemographic characteristics of the included participants with and without mental disorders are shown in Table 1. A large percentage of the participants with mental disorders were male and younger compared to those without mental disorders. Among the participants with mental disorders, a large percentage were single, experienced an economic impact, and had a household income level of less than 2 million JPY, with a small percentage having a household income level of more than 8 million JPY. With respect to the diagnosis of mental disorders, 46.3% of the participants were diagnosed with depression.

### 3.2. Changes in the PHQ-9 Score in Participants with and Without Mental Disorders

The results of the linear mixed model are shown in Table 2, and there was no significant interaction effect between survey time and the presence or absence of mental disorders with respect to the PHQ-9 score (*p* = 0.99). The main effect of survey time (*p* = 0.006) was significant, and the PHQ-9 score significantly decreased between Time 2 in January 2021 and Time 4 in January 2022 (*p* = 0.001). PHQ-9 scores were significantly higher among participants with mental disorders than among those without mental disorders.

### 3.3. Frequency of Use of Coping Strategies Among Participants with and Without Mental Disorders

The frequency of use of coping strategies at Time 5 is shown in Figure 3. The values of denial, substance use, use of instrumental support, behavioral disengagement, venting, humor, religion, and self-blame were not equally distributed between participants with and without mental disorders; therefore, Welch’s *t*-test was conducted.

Compared with participants without mental disorders, participants with mental disorders used 8 of 14 coping strategies more often: denial, substance use, the use of emotional support, the use of instrumental support, behavioral disengagement, venting, religion, and self-blaming. The other six coping strategies were not significantly different between participants with and without mental disorders.

### 3.4. Risk Factors for Depressive Symptoms in and Coping Strategies of Participants with and Without Mental Disorders by PHQ-9 Score

There was no correlation coefficient of 0.8 or higher, accordingly, all variables were included in the multiple linear regression analysis. Table 3 showed the results of the analysis of participants with and without mental disorders. There was no VIF of 5.0 or higher and adjusted r-squared values were adequate.

Among participants with mental disorders, being unmarried, use of emotional support, and behavioral disengagement were associated with higher PHQ-9 scores. There was no coping strategy associated with lower PHQ-9 scores.

Among participants without mental disorders, experiencing an economic impact from the COVID-19 pandemic, behavioral disengagement, and self-blaming were associated with higher PHQ-9 scores. Active coping and humor were associated with lower PHQ-9 scores.

## 4. Discussion

Previous studies have reported an increase in depressive symptoms following the onset of the COVID-19 pandemic in 2020 [1,2,3]. However, our study found that these symptoms decreased by January 2022, regardless of whether participants had mental disorders. Although participants with mental disorders employed certain coping strategies more frequently than those without, they did not engage in coping strategies that were adaptive for managing depressive symptoms.

With respect to changes in depressive symptoms, Romanyukha et al. [25] conducted a survey in Moscow, Russia, between September 2020 and December 2021, which included a lockdown period. Their findings indicated that the lockdown increased the daily risk of developing new cases of major depressive disorder. In Japan, the first mild lockdown occurred between April 2020 and May 2020; it was conducted intermittently and ended in September 2021. Thus, the improvement in depressive symptoms—regardless of mental disorder status—may be attributed to the lifting of the lockdown, and depressive symptoms could be expected to decrease in the long term. These findings suggest that for individuals with mental disorders, pandemic-related depressive symptoms may be temporary and more influenced by environmental factors than by coping ability. However, a previous study noted that major depressive episodes among the youth population remained elevated during the pandemic period compared with the pre-pandemic period [26], and further research is needed to consider the influence of age and attributes on changes in depressive symptoms.

With respect to coping strategy, participants with mental disorders used certain coping strategies more frequently to cope with the social situation during the COVID-19 pandemic than did participants without mental disorders; however, none of the coping strategies appeared to be effective in managing depressive symptoms. A previous study revealed that people with mental disorders tended to use passive and avoidance strategies to cope with the COVID-19 pandemic, such as denial, venting, substance use, behavioral disengagement, and self-blaming, and used less positive reframing, acceptance, emotional support, instrumental support, and self-distraction [4,12]. These findings are generally consistent with the present study’s results.

However, with respect to emotional support in the present study, the participants with mental disorders received more emotional support than those without; nonetheless, emotional support was positively associated with depressive symptoms in these individuals. A meta-analysis revealed that social support was weakly associated with mental disorder symptoms during the COVID-19 pandemic [27]. It is possible that individuals with more severe depressive symptoms tend to receive greater emotional support which may, in turn, heighten their awareness of their vulnerable state. In contrast, a systematic review revealed that social support was important for maintaining mental health during the pandemic [5]; accordingly, it is necessary to verify the concrete support and circumstances of receiving support and to reveal whether emotional support is effective or ineffective.

Additionally, marital status in 2020 was associated with depressive symptoms in 2022 among participants with mental disorders. In previous studies, there were two results concerning the correlation between marital status and mental health: one was that family support might be important [28,29,30], and the other was that having a family increases anxiety during the pandemic (Kabir et al., 2023 [31]; Zheng et al., 2024 [32]). According to the former studies, a meta-analysis found that loneliness was moderately associated with depressive symptoms during the COVID-19 pandemic [5,27]. According to the latter studies, married individuals in the Indian subcontinent were more likely to be anxious during the COVID-19 pandemic [31], and marital status among university students was correlated with depression [32]. Katchamart et al. [33] noted that there might be correlations among participants’ beliefs, religions, ethnicities, and socioeconomic structures.

Behavioral disengagement showed a positive association with depressive symptoms among participants with and without mental disorders. Behavioral disengagement refers to withdrawing from efforts to manage the situation; for example, avoiding information-seeking, decision-making, or adopting appropriate behavioral responses.

Therefore, in Japan, although emotional support is not always effective for treating depressive symptoms in people with mental disorders, the presence of a marital relationship is important for these individuals, and it is important to provide behavioral guidelines regardless of the presence or absence of mental disorders.

There were fewer coping strategies related to depressive symptoms among individuals with mental disorders. Effective coping has been reported to depend not only on individual characteristics but also on the degree of congruence with the stressor [34]. In the uncertain context of the COVID-19 pandemic, it was difficult for both individuals with and without disabilities to determine appropriate coping methods. In particular, at Time 5 in 2022, which was the focus of this study, the mild lockdown had been lifted in Japan, and people were able to engage in a wider range of thoughts and behaviors regarding infection prevention. Therefore, effective or ineffective coping styles for depressive symptoms during this period may have been intricately influenced by factors such as individual beliefs about infection prevention, which were not fully captured in this study. Another possibility is that the instruction for the Brief-COPE inventory specified ‘the frequency of coping methods used to handle social disruptions and challenges arising from the COVID-19 pandemic’, which may have limited the scope of coping strategies assessed.

The present study has some limitations. First, there was an issue with the sample sizes between participants with and without mental disorders. Although we ensured balanced distribution between participants with and without mental disorders to address the issue, Peduzzi, Concato, Kemper, Holford, and Feinstein [23] pointed out that the sample size needed to be 10 times the number of explanatory variables, the present study required 220 participants with mental disorders. Accordingly, it should be noted that results for participants with mental disorders may be affected by Type II errors, whereas results for those without may be subject to Type I errors.

Second, we did not assess specific diagnoses or symptom severity, as our aim was to identify generalizable trends and contribute to a broader understanding of individuals with mental health challenges. However, previous studies reported that the influence of the pandemic might vary depending on the type of disorder, such as anorexia nervosa, PTSD, and depression [6,35], and the use of coping strategies varies depending on the severity of the disease [12]. The findings should be interpreted with caution regarding their generalizability. In the future, it might be important to examine the characteristics of each specific diagnosis.

Third, although the results of the present study revealed that depressive symptoms decreased until 2022, it was unclear what kind of treatment the participants with mental disorders were receiving and whether depressive symptoms became more severe after the COVID-19 pandemic than before the pandemic.

Fourth, as this study utilized a web-based survey with self-reported questionnaires, there might have been self-selection and sampling biases among the participants. While 9.3% of participants in this study reported having a mental disorder, the 12-month prevalence of mental disorders in Japan has been reported as 5.2%, and the lifetime prevalence as 22.9% [36]. These discrepancies should be interpreted with caution, as they might be influenced by the survey methodology.

Fifth, approximately half of the participants who answered the survey at Time 0 dropped out of the study, and data were collected exclusively through a web-based survey. Population bias is thought to have occurred, and depressive symptoms measured by the PHQ-9 may not fully correspond to clinically diagnosed depression.

Finally, this study was unable to assess participants’ COVID-19 infection status or vaccination history, as the timing and accessibility of vaccination in Japan varied considerably depending on factors such as age, occupation, and the presence or absence of underlying health conditions. These limitations warrant further investigation.

## 5. Conclusions

The findings of the present study suggest that depressive symptoms associated with the COVID-19 pandemic might have diminished one to two years after the onset of the pandemic, regardless of whether people had mental disorders. This improvement might be partially attributed to environmental changes, such as the easing or lifting of lockdown measures. Although people with mental disorders reported a higher frequency of using coping strategies in response to the social disruptions caused by the pandemic compared to those without mental disorders, these strategies were not necessarily effective in reducing depressive symptoms. In particular, emotional support was more frequently utilized by people with mental disorders; however, its effectiveness appeared to vary depending on the content and context in which it was received. Notably, marital relationships were associated with lower levels of depressive symptoms among people with mental disorders, suggesting that close personal relationships might serve as a protective factor. Furthermore, the present study highlighted the importance of behavioral guidelines for managing the psychosocial impacts of the pandemic, regardless of mental health status. Overall, these results offer valuable insights that may inform strategies to support mental well-being during prolonged public health crises, particularly for those vulnerable to psychological stress.

## Figures and Tables

**Figure 1 healthcare-13-01095-f001:**
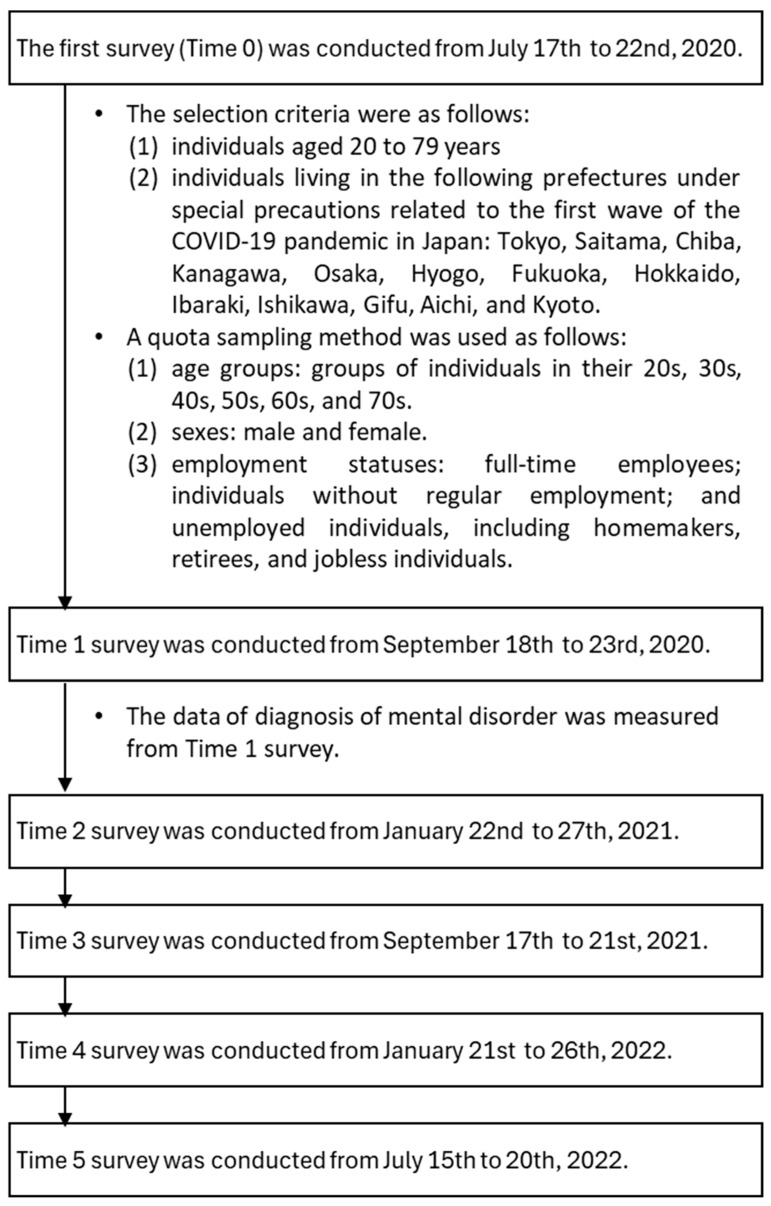
Procedure of the present study.

**Figure 2 healthcare-13-01095-f002:**
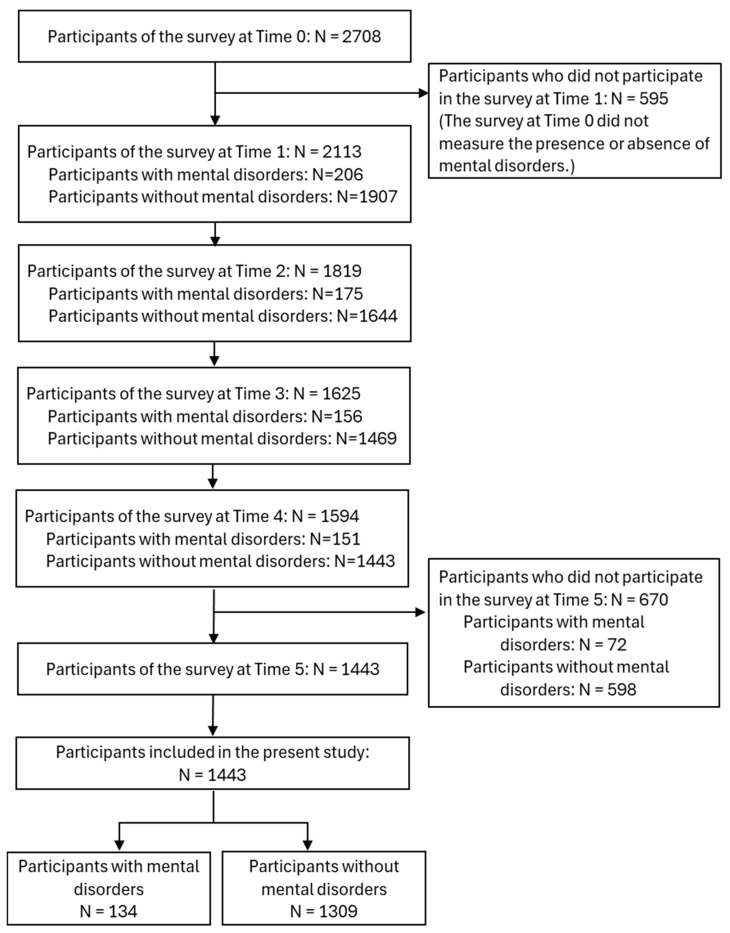
Flow chart of participant inclusion in the present study.

**Figure 3 healthcare-13-01095-f003:**
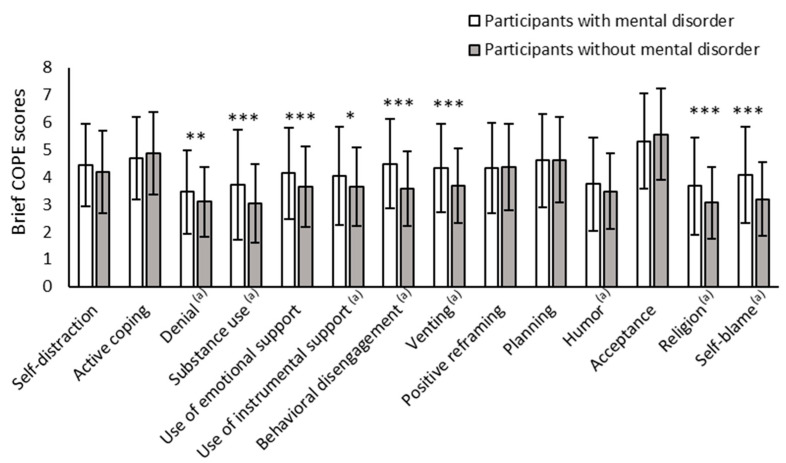
Mean scores and standard deviations of the Brief-COPE Inventory score at Time 5. * *p* < 0.05, ** *p* < 0.01, *** *p* < 0.001. (a) The values were not equally distributed between participants with and without mental disorders; therefore, Welch’s *t*-test was conducted.

**Table 1 healthcare-13-01095-t001:** Sociodemographic characteristics at Time 1.

	Participants with Mental Disorders		Participants Without Mental Disorders		Statistics
	N = 134		N = 1309		
	N/Mean	%/SD	N/Mean	%/SD	*p*
Sex (N/%)					
Male	90	67.2	715	54.6	0.005
Female	44	32.8	594	45.4	
Age (Mean/SD)	44.23	12.47	54.15	15.45	<0.001
Marital status (N/%)					
Single	92	68.7	540	41.3	<0.001
Married	42	31.3	769	58.7	
Household income level (N/%)					
Less than 2 million JPY	27	20.1	134	10.2	<0.001
Between 2 and 8 million JPY	71	53.0	707	54.0	
More than 8 million JPY	7	5.2	222	17.0	
Does not know or no answer	29	21.6	246	18.8	
Experienced an economic impact (N/%)	59	44.0	458	35.0	0.038
Diagnosis (N/%) ^(a)^					
Depression	62	46.3			
Stress-related and somatoform disorders	20	14.9			
Bipolar disorder	14	10.4			
Schizophrenia, schizotypal and delusional disorders	16	11.9			
Neurotic disorders	10	7.5			
Major neurocognitive disorder	4	3.0			
Other disorders	12	9.0			
Does not know disease name	35	26.1			

SD: Standard Deviation. ^(a)^ Multiple answers allowed.

**Table 2 healthcare-13-01095-t002:** Mean PHQ-9 scores estimated via a linear mixed model.

	Total		Participants with Mental Disorders		Participants Without Mental Disorders		*p*-Value		
	Estimate	SE	Estimate	SE	Estimate	SE	Interaction	Survey Time	With/Without Mental Disorders
Time 1	8.10	0.22	12.13	0.43	4.08	0.14	0.099	0.006	<0.001
Time 2	8.26	0.22	12.59	0.43	3.94	0.14			
Time 3	8.07	0.23	12.20	0.43	3.94	0.14			
Time 4	7.69	0.23	11.68	0.43	3.71	0.14			
Time 5	7.85	0.22	11.75	0.43	3.95	0.14			

SE: standard error.

**Table 3 healthcare-13-01095-t003:** Multiple linear regression for the PHQ-9 score at Time 5.

	Participants with Mental Disorders			Participants Without Mental Disorders		
	β	*p*	VIF	β	*p*	VIF
PHQ-9 score at Time 1	0.55	<0.001	1.35	0.62	<0.001	1.29
Sex	−0.04	0.571	1.14	0.01	0.450	1.39
Age	0.04	0.597	1.40	−0.04	0.075	1.39
Marital status	−0.17	0.049	1.44	−0.01	0.709	1.66
Household income level: Between 2 and 8 million JPY (Reference)						
Less than 2 million JPY	0.00	0.959	1.21	0.02	0.270	1.40
More than 8 million JPY	0.11	0.158	1.13	0.00	0.849	1.33
No answer or does not know	0.03	0.641	1.15	−0.04	0.056	1.22
Economic impact	−0.03	0.709	1.04	0.04	0.039	1.20
Brief-COPE Inventory score at Time 5						
Self-distraction	−0.05	0.649	2.05	0.04	0.114	2.21
Active coping	0.02	0.852	2.56	−0.10	0.001	2.72
Denial	−0.14	0.155	2.16	−0.02	0.486	2.06
Substance use	0.03	0.740	1.59	0.04	0.067	1.83
Use of emotional support	0.28	0.027	3.11	−0.02	0.457	3.45
Use of instrumental support	−0.05	0.656	2.74	−0.01	0.766	3.12
Behavioral disengagement	0.20	0.025	1.93	0.11	<0.001	1.82
Venting	0.01	0.959	2.13	0.04	0.131	3.50
Positive reframing	−0.05	0.651	2.44	0.05	0.103	3.04
Planning	−0.08	0.539	2.54	−0.05	0.073	3.56
Humor	0.02	0.842	1.69	−0.05	0.037	1.89
Acceptance	−0.06	0.495	1.78	0.05	0.052	1.90
Religion	0.01	0.910	1.76	0.01	0.658	2.43
Self-blaming	0.11	0.291	2.38	0.10	<0.001	2.25
Adjusted R^2^	0.40	<0.001		0.55	<0.001	

VIF: variance inflation factor.

## Data Availability

The datasets generated during the current study are available in the openICPSR database at https://www.openicpsr.org/openicpsr/project/211001/version/V2/view (accessed on 5 May 2025).

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
