# Peer review of "Depressive Symptoms and Risk Factors During the COVID-19 Pandemic Among People with/Without Mental Disorders"

_healthcare, 2025, doi:10.3390/healthcare13101095_

Round 1

Reviewer 1 Report

Comments and Suggestions for Authors

This study presents a longitudinal investigation into the evolution of depressive symptoms among individuals with and without pre-existing psychiatric disorders during the COVID-19 pandemic. By employing well-established psychological instruments and statistical analyses, the authors explore differences in symptom trajectories and coping mechanisms, providing valuable insights into mental health resilience and vulnerability in the context of a global crisis.

Strengths:

  1. Timely and Relevant Topic:
    The psychological impact of the COVID-19 pandemic remains a critical area of inquiry, particularly in populations with pre-existing mental health conditions. This study's focus on longitudinal trends adds important nuance to a field often dominated by cross-sectional designs.

  2. Robust Methodology:
    The use of the Patient Health Questionnaire-9 (PHQ-9) for depressive symptoms and the Brief COPE inventory for coping strategies ensures methodological rigor. Additionally, the application of linear mixed models and multiple regression analyses allows for a comprehensive statistical examination of both within-group and between-group differences.

  3. Key Findings with Practical Implications:
    One noteworthy result is the decrease in depressive symptoms over time regardless of psychiatric history, which suggests a form of psychological adaptation or recovery. The associations between emotional support, behavioral disengagement, and depressive symptoms offer actionable directions for mental health interventions.

Suggestions for Improvement:

  1. Clarify the Nature of Psychiatric Diagnoses:
    The paper would benefit from specifying which psychiatric disorders were included in the “mental disorder” category. Grouping diverse conditions together (e.g., anxiety, schizophrenia, bipolar disorder) might obscure meaningful differences in symptom progression and coping responses.

  2. Expand on Coping Strategy Findings:
    While the results mention that coping strategies were used more frequently by those with mental disorders, the manuscript could delve deeper into which specific coping styles were employed (e.g., avoidant vs. adaptive) and how these relate to the outcome measures.

  3. Address Limitations of Self-Report and Sampling:
    Given the web-based survey design, potential biases—such as self-selection and reliance on self-reported diagnoses—should be more thoroughly acknowledged. Representativeness and generalizability of the sample are also critical to discuss, particularly as only 9.3% reported having a mental disorder.

  4. Interpretation of Non-significant Results:
    The finding that no coping strategy significantly influenced PHQ-9 scores among those with mental disorders could be explored more critically. Is this due to a ceiling effect in symptom severity, or might certain coping styles have indirect or delayed effects not captured by the PHQ-9 alone?

  5. Language and Clarity Improvements:
    The manuscript would benefit from careful language editing to improve clarity. For example, the sentence “It might be important that a presence of a spousal relationship…” could be revised to “The presence of a spousal relationship may play an important protective role…”

Reviewer 2 Report

Comments and Suggestions for Authors

Dear Editorial Team

Thank you for the invitation.

This manuscript has interesting points, however, requires substantial modifications before consideration.

Firstly, the English language must be improved and the way of presentation needs more attention to conclude each finding at the associated section. Otherwise, it will not be interesting to read for readers.

Then, the protocol of the survey and the associated database, followed steps, including and excluding criteria must be clarified and if they are too long, they should be uploaded as supplementary files and the applied references for each protocol and data bank must be cited.

In the method section "the procedure" is difficult to understand. Is needs a graphical summary to shoe the timing and the criteria.

The applied abbreviations in the tables and text must be defined.

How did the authors prove the detected disorders?

In fig.2 the exact and detail description of each disorder must be clarified and the associated references must be cited.

The possible impact of COVID-19 and vaccination is not clearly discussed.

Authors must refer to long-term disorders following COVID-19 by comparing other studies (could be find at https://doi.org/10.1177/21501319241251941, doi: 10.3390/jcm12072673)

Abstract should transfer the main finding more efficiently.

The conclusion needs to be matched with the key findings. 

Round 2

Reviewer 2 Report

Comments and Suggestions for Authors

Dear Editorial Team

The authors have made fundamental improvement to the manuscript and it is more practical now.

The revised version is satisfying.

Best regards.

Comments on the Quality of English Language

Dear Editorial Team

The authors have made fundamental improvement to the manuscript and it is more practical now.

The revised version is satisfying.

Best regards.